# Seasonality and Small Spatial-Scale Variation of Chlorophyll *a* Fluorescence in Bryophyte *Syntrichia ruralis* [Hedw.] in Semi-Arid Sandy Grassland, Hungary

**DOI:** 10.3390/plants9010092

**Published:** 2020-01-11

**Authors:** Zsolt Csintalan, Evelin Ramóna Péli

**Affiliations:** 1Doctoral School of Biological Sciences, Institute of Botany and Ecophysiology, Szent István University, Páter Károly utca 1., H-2100 Gödöllő, Hungary; csintalan.zsolt@gmail.com; 2Department of Botany, University of Veterinary Medicine, István utca. 2., H-1078 Budapest, Hungary

**Keywords:** desiccation tolerance, dune, photochemical quenching, non-photochemical quenching, poikilohydric, maximum photochemical efficiency of photosystem II, photochemical quantum yield of photosystem II, non-photochemical quenching

## Abstract

Bryophytes face challenges due to global climate change which is leading to in-depth research in monitoring and studying their photosynthetic activity. The aim of this preliminary experiment was to study the seasonal variation trend in the chlorophyll *a* fluorescence parameters, Fv/Fm (ratio of variable to maximum fluorescence), photochemical fluorescence quenching (qP), photochemical quantum yield of photosystem II (ΦPS II), fluorescence quenching (qN), and non-photochemical quenching (NPQ), in the moss cushions of *Syntrichia ruralis* [Hedw.] collected from semi-arid sandy dunes for two slopes i.e., north-east (NE) and south-west (SW) direction. Our results showed a seasonal and small-spatial scale variation trend in all chlorophyll fluorescence parameters. These variations are due to different seasonal conditions referring to different degrees of environmental stress. ΦPS II and qP values were maximum in winter and in spring seasons while Fv/Fm, NPQ and qN were maximum in summer. Based on the different exposition of dunes, the SW slope showed increased values of the effective quantum yield of PS II and qP in comparison to the NE slope due to the optimal microclimate conditions for their expansion. These results may refer to the future changing in diversification and coverage of the *Syntrichia* species in semi-arid sandy grassland due to more effective metabolism in the beneficial microclimatic conditions.

## 1. Introduction

Cryptogamic species (such as algae, fungi, lichens, and bryophytes) collectively form biological soil crusts that perform an important ecological function i.e., production in dry grasslands [1]. Bryophytes are facing challenges due to climate change as their photosynthetic and biochemical activities primarily depend on the external environmental conditions [2]. Some of them have a special mechanism to protect the photosynthetic system from excessive light that allows them to grow successfully in a changing environment and different microhabitats.

Desiccation tolerance is a relatively common phenomenon among lichens and bryophytes, which helps in their survival under water-deficit conditions by recovering their metabolic activity upon rehydration. Mosses are poikilohydric and desiccation-tolerant ones such as *Syntrichia ruralis* [Hedw.] belong to the homoiochlorophyllous (HDT) groups [3] and cannot maintain constant internal water content by regulating water loss [4]. Previously, few studies have been undertaken at a physiological and molecular level to understand the mechanism behind the desiccation tolerance in bryophytes were reported that *Tortula ruralis* (=*Syntrichia*) are adapted and abundant in the semi-arid habitats and have the ability to tolerate high irradiation and temperature fluctuations [5]. ‘Blackspots’ of desiccated moss carpet of *S. ruralis* are visible in summer. In addition, it contributes 18–20% to the total cover found between the scattered tufts of dominant grasses *(Festucetum vaginatae danubiale* association) and plays an important role in the function of this community [6]. In hot and dry areas, due to a shortage of water, bryophytes adapt various survival strategies and grow in the more protected microhabitats such as at the bases of grass tussocks, on tree trunks and in rock crevices. Previously few experiments were conducted to study the photosynthetic behavior of *S. ruralis* moss after re-moistening in time [7]. In the dry state, they remained unchanged but after rehydration, they regained fully and rapidly their photosynthetically active state. These plants can stay in the desiccated state for months or years without dying then upon rehydration are able to recover to their full metabolic activity within minutes, hours, or a few days [8].

Bryophytes are the earliest group of first land plants that faced extremely dry conditions during movement to the terrestrial habitat. There has been limited information about their special adaptation strategy on a small spatial scale referring to chlorophyll fluorescence of desiccation tolerance. Only a few studies have been done on the impact of climate change on desiccation tolerant (DT) bryophytes. Global climate change causes high temperature, alternation in precipitation distribution and rising atmospheric CO_2_ which increases the demand for research to monitor and study their photosynthetic activity in the field of ecophysiology. Therefore, chlorophyll fluorescence measurements were examined as an indicator to understand how mosses respond to environmental changes.

In this study, chlorophyll *a* fluorescence was measured to study the recovery of photosynthetic activity of desiccant tolerant moss *S. ruralis* after rehydration in different slopes and seasons. Our work contributes to study the effect of small-spatial scale microhabitats with respect to seasons and provides information about the physiological adaptability of *S. ruralis* in a challenging environment. The hypothesis of the study was that in various exposures, the metabolic activity of *Syntrichia* species is different even in a relatively small-spatial scale patch of the microhabitat. We suppose that due to the climatic change, the composition, role, and production of one of the main biological crust components (*Syntrichia ruralis*) of the semi-arid sandy dry grassland might face significant changes in the future. Therefore, investigation on a small scale of seasonal variation of metabolic activity of *Syntrichia* sp. can give important information concerning the effects of future climate changes with a desertification aspect.

## 2. Results

### 2.1. Effect of Slopes (NE and SW) on Chlorophyll a Fluorescence Parameters

Chlorophyll fluorescence measurements were taken on desiccation-tolerant bryophyte *S. ruralis* showed recovery of Fv/Fm (ratio of variable to maximum fluorescence) within three days in the rehydrated state between NE and SW slopes respectively. An independent sample *t*-test was performed to calculate the significant values of two slopes with respect to fluorescence parameters. We found that there was no significant difference in Fv/Fm (*t*-test = −0.25922, df = 46, *p*-value = 0.7966). However, effective photochemical quantum yield of photosystem II (ΦPS II, *t*-test = −3.844, df = 46, *p*-value = 0.0003699) and coefficient of photochemical fluorescence quenching (qP, *t*-test = −4.1987, df = 46, *p*-value = 0.0001217) differed significantly between the two slopes. Non-photochemical fluorescence quenching (qN) parameters values were also significant (*t*-test = 2.5481, df = 46, *p*-value = 0.01424) and non-photochemical quenching (NPQ, *t*-test = 2.4668, df = 46, *p*-value = 0.01742) between NE and SW slopes (*p*-value ≤ 0.05) respectively. The photochemical fluorescence quenching parameters (ΦPS II, qP) showed higher mean values in the SW slope whereas non-photochemical fluorescence quenching parameters (qN, NPQ) increased in NE slope and Fv/Fm values were found to be similar mean values (Table 1).

### 2.2. Effect of Seasons on Chlorophyll a Fluorescence Parameters

One-way analysis of variance (ANOVA) was performed to determine the significant values of fluorescence parameters with respect to different seasons. ΦPS II has no significant difference whereas Fv/Fm, qN, NPQ values were found to be statistically significant within each pair of seasons as shown in (Table 2).

Seasonal variations were observed in all the fluorescence parameters values indicated different level of stress in *S. ruralis* is shown in (Figure 1). The highest value of maximum photochemical efficiency of photosystem II (Fv/Fm) was observed during hot and dry summer season (July) followed by late winter (March) then autumn season (October) and the lowest value was observed in the spring season (May). The photochemical quantum yield of photosystem II (ΦPS II) showed maximum value in the winter, followed by autumn, the spring and the minimum values in the summer season. The coefficient of photochemical fluorescence quenching (qP) is observed to be higher in the spring season followed by summer than winter and the lowest values are in the autumn season. Coefficient of non-photochemical fluorescence quenching (qN) and non-photochemical quenching (NPQ) showed similar trends and both have maximum values in the summer season followed by spring and winter season and minimum values in the autumn season (Figure 1).

### 2.3. Effect of Slopes and Seasons Interaction on Chlorophyll a Fluorescence Parameters

Multivariate Analysis of Variance (MANOVA) results were shown in (Table 3) there were no significant differences (*p* ≥ 0.05) in photochemical quenching parameters (Fv/Fm, ΦPS II, qP) values. However, non-photochemical quenching parameters (qN and NPQ) have a significant difference with respect to slopes within seasons. Chlorophyll fluorescence parameters measured during different seasons showed a *p* ≤ 0.05 expect for ΦPS II which may indicate seasonal variations.

## 3. Discussion

In this preliminary study, we analyzed the photosynthetic efficiency by measuring chlorophyll *a* fluorescence of desiccation-tolerant bryophyte *S. ruralis* within different slopes north-east (NE) and south-west (SW) during different seasons in semi-arid sandy grassland. The study revealed that ΦPS II, qP, qN, NPQ chlorophyll *a* fluorescence parameters showed small-spatial scale variations that were statistically significant between both slopes and seasonal variations were found to be clearly significant in the Fv/Fm, qN and NPQ fluorescence parameters (Table 3) when comparing *p*-values (≤0.05). These seasonal variations might be due to different environmental conditions such as irregular precipitation patterns, temporal distribution in xeric habitats [9,10]. Climate change causes a significant increase in temperature and a decrease in precipitation in summer for the Kiskunság sand ridge [11] and makes this area vulnerable to drought [12].

### 3.1. Photochemical Quenching Parameters (Ratio of Variable to Maximum Fluorescence (Fv/Fm), Photochemical Quantum Yield of Photosystem II (ΦPS II), Photochemical Fluorescence Quenching (qP))

In our results, Fv/Fm values showed a significant difference in summer and spring compared to winter and autumn (Table 2). The mean value of Fv/Fm was found low in the spring season which might be due to minimum temperature value (below 0 °C) in March 2018 (Figure 2A). In this period, sampling sites were covered with snow as the effect of a late winter and as a result delayed the recovery of photosynthetic activity of moss cushions that might have been due to cold temperature stress. Fv/Fm values were observed higher in both slopes in the summer period (Figure 1) which may indicate the effect of environmental stress conditions. In too hot and dry conditions, mosses become dormant because they have a short active period in the morning hours to activate photosynthetically when moisture is available [6,13]. Mosses experienced high irradiance and high temperature, as a result, they became inactive in the desiccated state [14].

In the autumn and winter season, Fv/Fm values were found to be mostly like each other. During colder conditions, mosses assimilate effectively and can continue their growth in a hydrated state under moderate temperature and irradiance except for snow cover period [14].

ΦPS II is the most useful parameter in photochemical quenching processes which measures the efficiency of PS II photochemistry and as an indication of overall photosynthesis and relates to achieved efficiency [15]. ΦPS II was observed to be highest in winter followed with little variation in autumn and spring seasons whereas lower values were found in the hot and dry summer and reflected the least photochemical efficiency [14]. Another similar parameter is qP with ΦPS II that gives an indication of the proportion of open PS II centers and relates to altered efficiency [16]. In Figure 1, qP values are shown to be maximum in the spring season.

In spring and winter season, we observed higher values for qP and ΦPS II parameters, respectively in the SW slope (Figure 1) which indicated the higher photosynthetic efficiency in the SW slope as compared to NE slope due to better optimal conditions such as longer favourable light conditions and availability of water which might allowed a longer active period for moss cushions in the SW slope on the dunes of semi-arid sandy grassland.

### 3.2. Non-Photochemical Quenching Parameters (Fluorescence Quenching (qN), Non-Photochemical Quenching (NPQ))

Non-photochemical processes for energy dispersion in the PSII act as mechanisms to reduce photoinhibition during a period of high light intensity [16]. On comparing the seasons, higher values of these parameters were recorded in the summertime (Figure 1 and Table 2). *S. ruralis* belongs to sun-exposed habitats that exhibit photoprotection at higher irradiance [17]. Similarly, when light is excessive, NPQ values become higher and indicate the protection of photosystem apparatus from excess excitation energy [18]. Therefore, qN and NPQ values were shown to be higher during the summer season.

In spring, there were little variations in qN and NPQ values which might be due to the transition from colder to warmer conditions. Also, in this season, temperatures increase rapidly, and mosses dry out in only a few hours after sunrise and become metabolically inactive due to high light exposure that causes photoinhibition [19]. Previous studies reported that rapid recovery of the photosynthetic system from dehydration allows it to react rapidly for short wet periods. This reaction due to the low water requirement of *S. ruralis* species enables it to survive during the hot and dry summer [6]. In an open sun-exposed habitat, high values of NPQ indicated water stress and high-light protection mechanisms [20]. In another moss species, *Atrichum androgyne* similar results were reported that NPQ values were increased during rehydration due to photoprotection and showed a high level of desiccation tolerance [21]. In the winter season, higher values were observed in qN and NPQ parameters in NE slope (Figure 1) which seemed to have resulted from a below-zero temperature which leads to cold temperature stress.

In this temperate climatic zone, spring, autumn, and winter generally have a favorable water supply but the summer season is strongly water limited. The main growing season in the open sandy grassland is the late spring in central and eastern Europe [6]. The small spatial scale microclimatic conditions may also modify the seasonal differences. The variations in chlorophyll fluorescence parameters observed were due to different environmental natural conditions in semi-arid sandy grassland in various seasons. Comparing the expositions, higher values were observed in NE slope for qN, NPQ except for Fv/Fm, ΦPS II, qP parameters and this might be due to fluctuations in some abiotic factors such as temperature, humidity, and light irradiance, soil parameters, precipitation, etc.

## 4. Materials and Methods

### 4.1. Sampling of Plant Material

*S. ruralis* (Hedw.) F. Weber and D. Mohr (synonymous: *Tortula ruralis*) belongs to family Pottiaceae and is known as sandhill screw moss. They are found in the form of extensive mats on open exposed areas of sandy dunes in semi-arid sandy grassland which plays an important role in binding sand particles. They are yellowish-green to golden brown in wet conditions and dark brown in dry conditions. Leaves are long with tapering tip end with silvery hair point. Samples were collected four times from semi-arid sandy grassland near Bócsa-Bugac in the Kiskunság region (central Hungary 46°53′29″ N, 19°26′35.6″ E) close to Bugac in dry state during late winter season (March 2018), spring (May 2018), summer (July 2018) and autumn (October 2018) from two different microhabitats north-east (NE) and south-west (SW) slopes on the basis of the orientation of sandy dunes and dominant wind direction.

### 4.2. Climatic Conditions

The average annual precipitation is 562 mm and the annual mean temperature is 10.4 °C. In the investigated year (2018), changes in the monthly average meteorological parameters (temperature, photosynthetically active radiation, precipitation, and relative humidity) at the Bugacpuszta site (46.69° N, 19.60° E; 111 ma.s.l.) in Figure 2A,B.

### 4.3. Experimental Set-Up

Dense and intact cushions of *S. ruralis* were collected in air-dried conditions and kept inside paper bags (16 × 13.5 cm) and at room temperature or 48 h in the open paper bag until the start of the experiment. These air-dried samples were cleaned and separated from the sand particles before conducting the experiment. After cleaning, these moss cushions were placed on a wet filter paper in a water-filled plastic box container (21.5 × 14 × 7 cm). Three moss cushions samples were placed in each plastic box container. They were sprayed with distilled water to maintain hydration for 3 days if necessary, till they reached a fully active rehydrated state under room temperature.

### 4.4. Chlorophyll a Fluorescence Measurement

Chlorophyll *a* fluorescence measurement was carried out on the moss cushions with a modulated chlorophyll fluorometer Hansatech Ltd. (King’s Lynn, UK) FMS 2. Calculation and definitions for chlorophyll fluorescence parameters (Fv/Fm, ΦPSII, qP) were followed as per [22] and (NPQ, qN) as per [23], respectively. The samples were maintained at a fully hydrated condition for 48 h at room temperature and placed it nearby the window. Prior to Fv/Fm measurements, the samples were kept in dark conditions for 30 min. This parameter has been widely used to measure the physiological condition of a plant in stress and estimates the maximum quantum efficiency of photosystem II. The values Fv/Fm for fully saturated, healthy and unstressed material are around in the range between 0.76 to 0.83 [23]. Measurements of 6 replicates were taken from each slope; NE and SW in four different seasons at room temperature. The samples were placed in the fluorometer and Fo (minimum fluorescence yield), Fm (maximum fluorescence yield) were recorded. The light intensity of the modulated measuring beam (1.6 kHz) was 100–150 nmol photons m^−2^ s^−1^, actinic light (650 nm, 370 µmol photons m^−2^ s^−1^) was used to assess steady-state fluorescence and the maximum fluorescence level was measured with saturating white light pulses of 3000 µmol photons m^−2^ s^−1^. The effective PSII quantum yield (ΦPSII), the potential quantum yield of PSII (Fv/Fm), the photochemical fluorescence quenching (qP), and the non-photochemical quenching (NPQ) were observed for dark-adapted samples using chlorophyll fluorometry method described by [16]. The protocol of analysis of chlorophyll fluorescence quenching, the calculation of fluorescence parameters and the standards are based on: Fv/Fm = (Fm − Fo)/Fm, ΦPSII = (Fm’ − Fs)/Fm’, qP = (Fm’ − Fs)/(Fm’ − Fo’), NPQ = Fm/Fm’ − 1.

### 4.5. Statistical Analysis

Experimental data were checked for normality with Shapiro–Wilk’s test and Levene’s test for equal variances. An independent sample t-test was performed to compare mean values of chlorophyll *a* fluorescence parameter between NE and SW slopes. A one-way ANOVA parametric test was used to check the significant differences in terms of fluorescence parameters (Fv/Fm, ΦPS II, qP, qN, and NPQ) between slopes (NE vs. SW slope) and season (spring, summer, autumn, winter). A multi-way ANOVA parametric test was done to study interaction within slope and seasons. ANOVA post-hoc (Tukey’s test) was performed at 95% confidence level to determine the significant differences between each pair of seasons with different parameters. Statistical analyses were performed using the statistical software R programming language version 3.5.3 for windows (R development Core Team, Auckland, New Zealand).

## 5. Conclusions

Our preliminary study showed clear and significant seasonal variations in chlorophyll *a* fluorescence parameters in desiccation-tolerant *S. ruralis* between NE and SW slopes located within a small distance from each other. Our results indicated better photosynthetic performance in the south-west slope (SW) in contrast to the north-east slope (NE) in all seasons. The presence of differences in photosynthetic properties in such a small-spatial scale of the microhabitat refers to the high adaptation ability and sensitivity level of mosses to the smallest changes in environmental factors. *Syntrichia ruralis* is one of the key components of the biological soil crust in semi-arid sandy grassland which are the main producers and relevant participants in the biological (mainly the carbon cycle) circulation of this ecosystem. Maximal utilization of better environmental factors (e.g., light) was shown in the higher activation of the photochemistry of SW slope species opposite to NE ones where the non-photochemical energy dissipation is more expressed. The cost of the more effective light absorption also attends to the need for greater water supply which was also limited in this habitat. Therefore, maintenance of the balance in photosynthesis (light reaction and carbon cycle) basically determined the spreading and production of mosses. Effects of changing the climate for the ecosystem especially for vegetation is more traceable by small spatial-scale investigation of such adaptable plants as mosses and can serve for prediction in the future.

## Figures and Tables

**Figure 1 plants-09-00092-f001:**
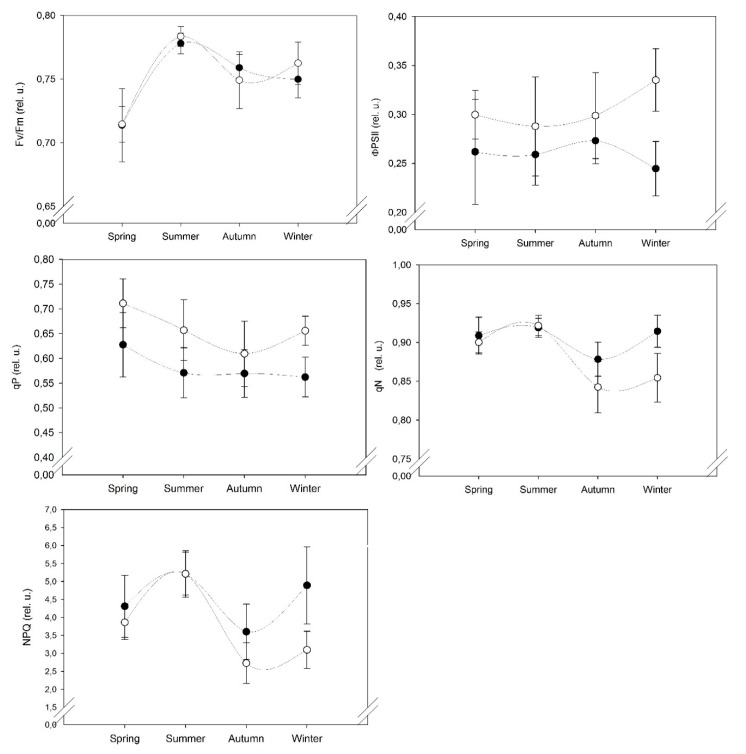
Mean of chlorophyll *a* fluorescence parameters (Fv/Fm, ΦPS II, qP, qN, NPQ) for *S. ruralis* were plotted against four seasons (Spring, Summer, Autumn, Winter) in two different microhabitats NE: north-east (●), SW: south-west (○) slopes. Error bars represent standard error of the mean.

**Figure 2 plants-09-00092-f002:**
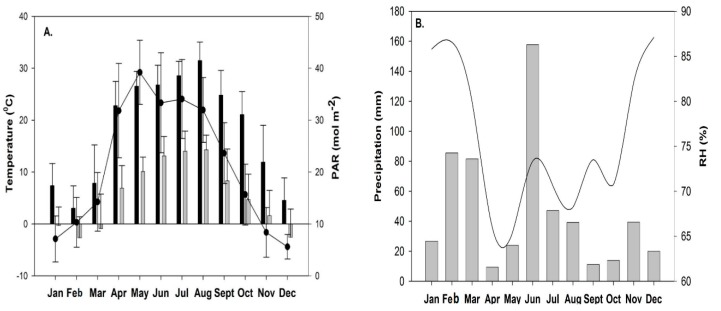
(**A**) Monthly average of the maximum and minimum values of air temperature (in 0 °C, Tmax, black bars and Tmin, grey bars) and the photosynthetically active radiation (PAR, in mol m^−2^) values at Bugac site. (**B**) Monthly sum of precipitation (mm) and average relative humidity (%) during the study period in 2018 at the Bugac site.

**Table 1 plants-09-00092-t001:** Chlorophyll *a* fluorescence parameter (Fv/Fm (ratio of variable to maximum fluorescence), photochemical quantum yield of photosystem II (ΦPS II), photochemical fluorescence quenching (qP), fluorescence quenching (qN), and non-photochemical quenching (NPQ) for *S. ruralis* after three days of recovery (fully hydrated state) at two different expositions of the dune (NE: north-east slope, SW: south-west slope).

Slopes	Fv/Fm	ΦPS II	qP	qN	NPQ
**North-east (NE)**	0.750 ± 0.006	0.260 ± 0.008	0.582 ± 0.012	0.905 ± 0.005	4.504 ± 0.217
**South-west (SW)**	0.752 ± 0.006	0.305 ± 0.009	0.659 ± 0.013	0.879 ± 0.008	3.724 ± 0.229

Data are expressed as mean ± standard error of six replicates.

**Table 2 plants-09-00092-t002:** Analysis of variance (ANOVA) results of chlorophyll a fluorescence parameter values (Fv/Fm, ΦPS II, qP, qN, NPQ) between slopes (NE and SW) with respect to different seasons (spring, summer, autumn, winter) in *S. ruralis*.

Season	Fv/Fm	ΦPS II	qP	qN	NPQ
Spring	0.714 ± 0.006 ^a^	0.280 ± 0.013 ^a^	0.670 ± 0.021 ^b^	0.904 ± 0.005 ^bc^	4.086 ± 0.022 ^a^
Summer	0.780 ± 0.002 ^c^	0.273 ± 0.013 ^a^	0.613 ± 0.021 ^ab^	0.920 ± 0.003 ^c^	5.212 ± 0.187 ^b^
Autumn	0.754 ± 0.005 ^b^	0.285 ± 0.011 ^a^	0.590 ± 0.018 ^a^	0.860 ± 0.010 ^a^	3.164 ± 0.243 ^a^
Winter	0.756 ± 0.005 ^b^	0.290 ± 0.016 ^a^	0.609 ± 0.017 ^ab^	0.884 ± 0.012 ^ab^	3.994 ± 0.370 ^a^

Data are expressed as mean ± standard error of six replicates from both microhabitats. Different alphabets (superscript) are represented a significant difference among all four seasons at *p* ≤ 0.05.

**Table 3 plants-09-00092-t003:** Multivariate Analysis of Variance (MANOVA) results of chlorophyll a fluorescence parameter values (Fv/Fm, ΦPS II, qP, qN, NPQ) between slopes (NE and SW) with respect to different seasons (Spring, Summer, Autumn, Winter) in *S. ruralis*.

Chlorophyll *a* Fluorescence Parameter	Variables	Sum of Squares	Df	F-Value	*p*-Value
**Fv/Fm**	Slopes	6.07 × 10^−5^	1	0.1806	0.6731
	Seasons	0.273314	3	27.0883	9.898 × 10^−10^ ***
	Slopes: Seasons	8.028 × 10^−4^	3	0.7956	0.5036
	Slopes	0.25025	1	14.7795	4.235 × 10^−4^ ***
**ΦPS II**	Seasons	1.830 × 10^−3^	3	0.3603	0.7819465
	Slopes: Seasons	8.346 × 10^−3^	3	1.643	0.1947031
	Slopes	0.69312	1	20.8218	4.709 × 10^−5^ ***
**qP**	Seasons	0.042304	3	4.2362	0.01084
	Slopes: Seasons	5.400 × 10^−3^	3	0.5408	0.65714
	Slopes	7.8541 × 10^−3^	1	12.8124	9.211 × 10^−4^ ***
**qN**	Seasons	0.0240497	3	13.0774	4.283 × 10^−6^ ***
	Slopes: Seasons	7.0756 × 10^−3^	3	3.8475	0.0164519 *
	Slopes	7.3039	1	11.9309	1.32 × 10^−3^ **
**NPQ**	Seasons	25.4975	3	13.8834	2.379 × 10^−6^ ***
	Slopes: Seasons	5.2303	3	2.8479	0.04949 *

*p*-values are expressed along with (*) indicated the different levels of statistical significance where (*, ** and *** represent *p* ≤ 0.01, 0.001 and 0.0001, respectively)

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
