# Peer review of "Seasonality and Small Spatial-Scale Variation of Chlorophyll a Fluorescence in Bryophyte Syntrichia ruralis [Hedw.] in Semi-Arid Sandy Grassland, Hungary"

_plants, 2020, doi:10.3390/plants9010092_

Round 1

Reviewer 1 Report

This version of the manuscript has been significantly improved. Authors now include information and clarification of some points. However,  one important info is still missing regarding the fluorescence measurements. The new sentence from line 245 to 250 is now clear for me, and yet I still don’t get which actinic light intensity they used to measure and calculate the parameters (ΦPS II, qP, qN, NPQ). I think that the mentioned sentence should be replaced for one simple sentence including the PAR of the actinic light and the time of illumination used for these determinations, that would be enough.

Author Response

Responses to Reviewer #1

Thank you for reading through the ms and for the valuable comments.

We looked through the manuscript carefully and revised the mentioned parts of the ms. We tried to simplify the marked sentence referring to chlorophyll fluorescence measurements according to your comments. We also rewrite and modify the English language and style.

Thank you very much for your comments, which greatly helped us in improving the manuscript.

GödöllÅ‘, 4th December 2019.

Sincerely Yours

on behalf of the authors

                                                                                                            Ruchika

Reviewer 2 Report

Some of the points indicated in the previous revision have been addressed, however important questions remain to be answered. The growth of the plants depend on multiple factors other than the photosynthetic parameters estimated in the work. It is not possible to predict the growth of a plant based solely in  the slight changes on the value of some photosynthetic parameters. Other parameters should be estimated such as CO2 fixation rate, carbon/nitrogen balance, etc…

In Abstract authors say that ΦII and Fv/Fm values do not change during the seasons, but at the same time they say that ΦII and qP values were maximum in winter. This is a serious inconsistency.

English should have been revised because it makes difficult the understanding of the manuscript

Author Response

Responses to Reviewer #2:

Thank you for reading through the ms and for the valuable comments.

Please find our answers below for your comments.

It’s absolutely true that we can’t conclude finally and fully conclusions about plants growing and possible future diversification in a certain habitat based on the solely chlorophyll fluorescence parameters. But this method is very sensitive to show the small differences from the plant physiological mainly the photosynthetically background mechanisms. Therefore, it is an adequate method to investigate the plants and especially the moss Eco physiological processes on a small spatial scale because they are responses faster and particularly prompt for the environmental changes than higher plants due to their more simply organization and indicator aspects. In the future, we are planning to measure the CO2 fixation rate and estimate the carbon balance of different microhabitats. We suppose to be a good correlation with chlorophyll fluorescent parameters. We modified the mentioned sentence in the Abstract, it was not so clear. Originally, we would like to refer to the differences in an insignificant level of the parameters. We also edited the English language and style.

Thank you very much for your comments, which greatly helped us in improving the manuscript.

GödöllÅ‘, 4th December 2019.

Sincerely Yours

on behalf of the authors

                                                                                                                 Ruchika

Reviewer 3 Report

The authors fulfilled the majority of the requirements in my 1st review. However, there are still some problematic issues.

It is still not clear what was the actinic light (PPFD) level used for determination of fluorescence parameters in the illuminated samples. From the part of the M&M where authors described measuring condition (lines 245 - 250) I would suppose that they utilized some pre-programmed protocol given by the instrument they used (a modulated chlorophyll fluorometer Hansatech Ltd., King ‘s Lynn,UK). However, the saturation pulse method allows determination of light response curves for certain parameter (e.g. qP) and for different PPFD's only by using valid protocol where chosen actinic light level (PPFD) has to be maintained until F and F'm reach steady values for at least 2-3 last saturation pulses. All other protocols are considered to be arbitrary and since they are not scientifically acceptable. Please describe this part of the methodology in appropriate way. The English language must be improved.

Author Response

Responses to Reviewer #3

Thank you for reading through the ms and for the valuable comments.

We looked through the manuscript carefully and revised the mentioned parts of the ms. We tried to simplify the marked sentence referring to chlorophyll fluorescence measurements according to your comments.We also improved the English language and style.

Thank you very much for your comments, which greatly helped us in improving the manuscript.

GödöllÅ‘, 4th December 2019.

Sincerely Yours

on behalf of the authors

                                                                                                               Ruchika

Round 2

Reviewer 1 Report

I have seen the changes operated in this reviewed version of the manuscript. The methods have been integrated and the operations performed in the manuscript are now explained sufficiently.

I think the manuscript is now suitable for publication.

Author Response

Thank you for reading through the ms and for the valuable comments.

Thank you very much for your comments, which greatly helped us in improving the manuscript and your suggestion for publication.

GödöllÅ‘, 24th December 2019

Sincerely Yours

on behalf of the authors

Ruchika

Reviewer 2 Report

No significant changes has been made with respect to the previous version. The main objection remain to be addressed: the growth of the plants depend on multiple factors. To understand the growth and development of the plant and to establish a relationship with external conditions, such as those proposed in the manuscript, other parameters should be estimated, such as CO2 fixation rate, carbon/nitrogen balance, etc…

Author Response

Responses to Reviewer #2:

Thank you for reading through the ms and for the valuable comments.

Please find our answers below for your comments.

The best method to be investigated the growth and development of the plants can be widely extended to measure many photosynthetic and production parameters such as CO2 fixation and carbon/nitrogen balance. These parameters can be played a very important role to reflect the significance of e.g. the mosses in the characteristic plant association of the investigated semi-arid area. These parameters relation with external environmental factors are mainly affected by climatic change. But our preliminary work tried to reveal the small differences in the working of one of the main components of the investigated ecosystem used chlorophyll fluorescence parameters in small-spatial scale, in different expositions. Finally, and primarily our aim is not to the large investigation from the degree of plant growing and development in a certain ecosystem but the detectable effects in small microhabitats.     

Thank you very much for your comments, which greatly helped us in improving the manuscript.

GödöllÅ‘, 24th December 2019

Sincerely Yours

on behalf of the authors

Ruchika

Reviewer 3 Report

Authors accepted all required remarks. So, I would reccomed the presen manuscript for publication.

Author Response

Responses to Reviewer #3

Thank you for reading through the ms and for the valuable comments.

Thank you very much for your comments, which greatly helped us in improving the manuscript and your recommendation for publication.

GödöllÅ‘, 24th December 2019

Sincerely Yours

on behalf of the authors

Ruchika

This manuscript is a resubmission of an earlier submission. The following is a list of the peer review reports and author responses from that submission.

Round 1

Reviewer 1 Report

In the present study authors investigated photosynthetic activity of desiccant tolerant moss S. ruralis grown in different slopes (north-east (NE) and south-west (SW)) and seasons (spring, summer, autumn and winter) after rehydration. The chapter Introduction is sufficiently informative concerning the chosen topic. Methods used in the present investigation are well chosen. Results are clearly presented and Discussion is mostly correctly written. However, there are certain objections that must be accepted:

1. Although the aims of the study are clearly stated and explained in the chapter Introduction the hypothesis is missing. Though the authors stressed that this is preliminary investigation, the hypothesis must be clearly stated at the end of this chapter.

2. In the chapter M&M:

- line 210: how long the plant material has been kept at room temperature before the experiment?

- line 215: how did you asses when a fully active rehydrated state was reached? Would the Fv/Fm value be a good indicator for this?

- line 223: the parameter F0 is not F and the letter ”o” but F and the number 0 in the subscript. Please correct this throughout the manuscript;

- line 225: please include the data on the photosynthetic photon flux density (PPFD)  by which the samples were illuminated during fluorescence measurements and explain why did you choose exactly this actinic light level;

- line 227: the coefficient of non-photochemical fluorescence quenching is usually written as qN (not qNP). So, please correct this throughout the manuscript and give the relevant reference for the calculation of all fluorescence parameters used in this investigation.

3. In the chapter Conclusion you have stated that your results: “.. indicated better photosynthetic performance in the Northeast slope (NE) in comparison to the Southwest slope (SW).“ However, data given in the Tables 1 and 3 as well as in Fig. 1 do not support your conclusion. The SW samples had increased values of the effective quantum yield of PSII and qP in comparison to the NE samples, what indicated their better capability to direct a greater fraction of absorbed light into the photosynthesis (this is photosynthetic efficiency or activity, not the NPQ!!!). Please adjust your conclusion (and discussion also) to your results as well as with the hypothesis that must be given in the chapter Introduction.

Reviewer 2 Report

In this work authors analyze some photosynthetic parameters in the bryophyte Syntrichia ruralis in a semi-arid sandy grassland ecosystem in Hungary. They determine the values of those parameters along the different seasons. I find that the main objection to this work is that they just determine the values of those photosynthetic parameters. There are not any other data to establish interconnections or relationships that allow to get some conclusion. There is not information about the growth conditions of the moss: temperature, light irradiance, humidity, length of the light period … In addition, the growth of the plants depends on multiple factors other than those photosynthetic parameters estimated in the work. What is the goal of authors with this work? Is it to use those parameters to predict the growth of this bryophyte? To determine which ecosystem or season is more suitable for the growth of the plant? I think in both cases other parameters should have been determined. Another important question: what is the novelty of this work? Considering that authors indicate that a previous work reported that spring and autumn were the main growth periods for S. ruralis in Hungary. Cyanobacteria are prokaryotes. They should not be considered as “cryptogamic species” (Introduction, lane 33). English should be revised thoroughly as it makes difficult the understanding of the manuscript

Reviewer 3 Report

The manuscript is kind of descriptive but has some interesr for ecological purposes. However, it needs extensive modification in order to, possibly, become publishable.

First, the wording/spelling and the English in general, I think a proper editing work is required to get some clarity. 

About the results, they seem ok, at least for the ecological approach they are interested in (they are clearly not physiologists or biophysicists), but the

data management needs also improvements. The way the results are presented is not easy to follow and the discussion is also a bit chaotic, maybe it

could be advantageous to get some help from someone more familiar with PAM measurements to improve data presentation, interpretation and conclusions.

In particular,  the material and methods section needs some clarifications: the most important is a detailed explanation of the protocol they used for

themeasurements (time of illumination, actinic light intensities, intensity and duration of the saturating pulse, …).

Finally, the statistical analysis would be better, for example in the multiple ANOVA analysis they should test the normal distribution of the residues and

lack of interactions between the factors, the two major assumptions of this parametric test.

To sum up, I think it can be published, because the data could be interesting to environmental and/or ecological public, but after an extensive editing

of the text and the way the results are presented and analysed.